# Improved Ultrasound-Guided Balloon-Assisted Maturation Angioplasty Using Drug-Eluting Balloons in the First Autogenous Arteriovenous Fistula Procedure: Early Experience

**DOI:** 10.3390/biomedicines12051005

**Published:** 2024-05-02

**Authors:** Domenico Mirabella, Ettore Dinoto, Edoardo Rodriquenz, Michele Bellomo, Andrea Miccichè, Paolo Annicchiarico, Felice Pecoraro

**Affiliations:** 1Vascular Surgery Unit, AOUP Policlinico “P. Giaccone”, 90127 Palermo, Italy; dmirabella@live.it (D.M.); edoardo.rod95@gmail.com (E.R.); bellomomichele1@gmail.com (M.B.); andrea.micciche01@gmail.com (A.M.); annicchiarico.paolo@gmail.com (P.A.); felice.pecoraro@unipa.it (F.P.); 2Department of Surgical, Oncological and Oral Sciences, University of Palermo, 90127 Palermo, Italy

**Keywords:** arterio-venous fistula, arteriovenous fistula maturation, early fistula failure

## Abstract

In patients with end-stage renal failure requiring hemodialysis, autogenous arteriovenous fistula (AVF) is preferred over tunneled dialysis catheters due to lower complications and costs. However, AVF maturation failure remains a common issue due to small vein size, multiple venipunctures, and other factors. Guidelines recommend using vessels of >2 mm for forearm AVFs and >3 mm for upper arm AVFs. This study investigates the use of intraoperative Doppler ultrasound (DUS)-guided Balloon-Assisted Maturation (BAM) with drug-eluting balloons (DEB) during initial AVF creation. Data from 114 AVF procedures, of which 27.2% underwent BAM, were analyzed. BAM was performed in 25 distal radio-cephalic and 6 proximal brachio-cephalic AVFs. With DUS guidance, vein stenosis was identified and treated using DEB. Technical success was achieved in all cases, with no early mortality. Early BAM-related complications were minimal, and no AVF thrombosis occurred. AVF maturation time was 15 days (SD: 3), and no further complications were reported during a mean follow-up of 10.38 months. Using BAM with DEB during AVF creation led to successful maturation and dialysis use without the need for secondary procedures. This study emphasizes the importance of identifying AVF failure risk early and utilizing DUS-guided procedures to enhance AVF outcomes. A more liberal use of intraoperative BAM could limit reinterventions in patients undergoing AVFs.

## 1. Introduction

AVF is the first choice in patients presenting end-stage renal failure requiring hemodialysis. Compared to tunneled dialysis catheters, AVF is associated with lower postoperative complications rate and costs with increased life expectancy and life quality [1,2]. Despite the many advantages of the native AVF, a number of studies have documented major problems with AVF maturation (failure to increase flow and diameter adequately to support dialysis) as a result of a peri-anastomotic venous segment stenosis due to small anatomic conformation, multiple venipunctures, history of phlebitis [3,4,5]. Clinical guidelines provide recommendations on the minimal blood vessel diameters required for arteriovenous fistula creation but the evidence for these recommendations is limited [6]. The ESVS Clinical Practice Guidelines suggest the use of arteries and veins > 2 mm for forearm fistulas and >3 mm for upper arm fistulas in order to obtain an AVF with an initial flow rate of 300–350 mL/min [7]. A good fistula should be easy to prepare, long lasting, free from complications, and aesthetically acceptable and economical. However, the anatomical and clinical variables of a patient may make it necessary to choose vessels with a smaller caliber, creating fistulas outside the classic recommendations.

Native AVF’s main drawback, and the consecutive association with dialysis unsuitability, is the lack of maturation occurring in up to 60% of patients at 4–5 months from the initial AVF creation [8]. The indicators of future AVF maturation failure are a thrill absence or reduction at 1 cm proximal to the AVF anastomosis and a reduced DUS volume flow [9]. These parameters, often assessable at the end of the first surgical procedure, can be corrected later with additional surgical or endovascular procedures. The Balloon-Assisted Maturation (BAM) is a common technique to improve AVF hemodynamics. Typically, BAM is carried out a second time through radial or brachial artery access, or via direct vein puncture using contrast agent. This procedure is commonly described as an additional step in follow-up care to enhance small vein diameter and address AVF maturation failure in patients [10]. However, BAM under duplex ultrasound (DUS) guidance is an alternative approach that reduces the complications related to a radiological procedure. Herein, we report the early experience with intraoperative DUS-guided BAM of autogenous AVF using drug-eluting balloons (DEB) during the first AVF performance. 

## 2. Materials and Methods

From January 2021 to March 2022, data from patients undergoing AVF creation were retrospectively analyzed and inserted into standard piloted forms. Patients treated with intraoperative BAM using DEB in correspondence with the efferent vein under DUS guidance are the study cohort. 

The indications of intraoperative BAM were intraoperative thrill absence, thrill reduction at 1 cm proximal to the AVF anastomosis, and a reduced DUS volume flow (<200 mL/min). Patients who had undergone diagnostic fistulography or other surgical procedures for execution of AVF in the same upper limb were excluded from analysis. The study was performed in agreement with the Declaration of Helsinki and STROBE guidelines for reporting observational studies were followed [11]. All patients gave informed consent for the procedure itself, anonymous data collection, and analysis. According to the National Policy in matter of Privacy Act on retrospective analysis of anonymized data, the local institutional review board approved the study. The retrospective and anonymized nature of this study did not require any medical ethical committee approval. Demographics, comorbidities, medical treatments, clinical presentations, and procedural data were collected. 

The procedures were carried out in the operation room equipped with a DUS (Siemens Acuson P500, Healthineers, Mountain View, CA, USA) in a standardized fashion. Baseline DUS investigation included complete arterial and vein assessment with flows, diameters, and cutaneous markers (Figure 1).

In patients with a history of heart failure or ejection fraction (EF) below 45%, a distal AVF was preferred to proximal site even in the presence of small caliber vessels to avoid pulmonary hypertension and acute pulmonary edema due to an increase in the vein flow [12].

AVF was carried out with a standard surgical technique under local anesthesia. When the chosen vein had a small diameter, a regional block anesthesia was chosen to exploit the known vasodilatory effect [13]. At AVF completion, clinical signs and DUS were evaluated intraoperatively. For patients presenting reduced/absent thrill and/or reduced DUS volume flow (<200 mL/min), a DUS to identify potential efferent vein stenosis was conducted (Figure 2).

In case of evident vein stenosis/reduced size, a BAM was performed through vascular access from the efferent vein, approximately 1 cm after the anastomosis (Figure 3A,B). Vessel stenosis was defined as at least a 50% decrease in the lumen diameter compared with the adjacent segment of the AVF vein.

A 5F introducer radial sheath (Radiofocus introducer transracial Kit, Terumo Corporation, Tokyo, Japan) was inserted in a retrograde fashion and the vein lesion was crossed using a 0.018 wire (Command; Abbott Vascular, Santa Clara, CA, USA) (Figure 3C). A DEB (Luminor; iVascular, Vascular, S.L.U., Barcelona, Spain) was positioned at the level of the lesion and inflated to its nominal pressure (Figure 4). 

The angioplasty time was 3 min, and the DEB diameter choice was 2 mm larger than the target vein, with a length equal to that measured during the ultrasound evaluation. After BAM, the efferent vein flaw was checked with DUS, and the thrill appearance was registered (Figure 5). 

After the AVF and BAM procedures, the administered heparin was not reversed, and a single prophylactic low molecular weight heparin (LMWH) shot was administered after 6 h from the index procedure. Therapy with a single antiplatelet agent was prescribed at the discharge. Technical success, early (within 30 days), and late (after 30 days) results were the measured outcomes. Technical success was defined as the intraoperative presence of thrill. AVF maturation was defined as the time from the index procedure to the first hemodialysis. Early and late outcomes included mortality, morbidity, vein thrombosis, and time or lack of AVF maturation. Follow-up consisted of clinical examination, AVF DUS on the first postoperative day, 1 and 6 months, and annually. 

Statistical analysis. Parametric data are presented as the mean and interquartile range (IQR) or median and min–max range; absolute values and percentages for non-parametric data. Differences in preoperative and postoperative outcomes were assessed using the Student *t*-test. Statistical significance was considered at *p* < 0.05. Statistical analysis was performed using SPSS 16.0 (SPSS Inc., Chicago, IL, USA).

## 3. Results

From the 114 AVFs carried out in the study period, 31 cases (27.2%) underwent BAM and were included in the study. Preoperative comorbidities and clinical findings are reported in Table 1. 

The intraoperative BAM was carried out in 25 distal radio-cephalic AVFs, and 6 proximal brachio-cephalic AVFs. The location of AVF was at the operator’s discretion after DUS examination. Recent vein punctures were reported in 27% of cephalic veins in correspondence with antecubital fossa; these were not considered as a limiting factor for AVF creation.

In radio-cephalic AVF, the mean transverse diameter of the afferent radial artery was 2.7 (IQR: 1.9–3.2; SD: 0.8) mm; and the mean transverse diameter of the efferent cephalic vein at the forearm was 2.2 (IQR: 1.8–3.0; SD: 0.7) mm. In brachio-cephalic AVFs, the mean transverse diameter of the afferent brachial artery was 5.1 (IQR: 4.0–6.1; SD: 0.9) mm; and the mean transverse diameter of the efferent cephalic vein at antecubital fossa was 3.1 (IQR: 2.5–4.8; SD: 0.8) mm.

Technical success was achieved in all cases, and no early mortality was registered. An early BAM-related complication was registered in three (9.7%) patients, including a DUS vein spasm in two patients, which resolved spontaneously within 24 h, and one case of bleeding from the sheath vein access, requiring surgical repair. No early AVF thrombosis occurred, and the registered maturation time, at DUS examination, was 15 (IQR: 12–19; SD: 3) days. During the mean follow-up of 10.38 (SD: 4) months, no additional complications were registered, all AVFs were used for dialysis and no secondary procedure was needed. 

## 4. Discussion

Autologous AVF is considered the first choice for hemodialysis access due to lower infection and reintervention rates [1]. However, the planning, execution, and maturation of a fistula are not trivial aspects. Using a vein as distal as possible, to facilitate punctures and make the puncturable vein section longer, carries a higher risk of AVF failure due to possible complications arising from an anastomosis created with small caliber vessels. Previous research has shown that using small diameter veins for access is directly related to the number of revisions and complications with a consecutive increased number of failure fistulas [14,15]. Allon and Robbin reported veins < 2.5 mm in diameter to have poor outcomes, and recommendations for not using smaller-diameter veins have been established [16]. The failure rate is estimated at 20–60% and the main causes are stenosis or expansion failure, resulting in an insufficient flow rate for dialysis use. The available evidence, based mainly on moderate-quality randomized controlled trials, suggests that the preoperative clinical examination should always be supplemented with routine DUS mapping before AVF creation. This policy effectively prevents unnecessary surgical explorations and greatly reduces the immediate AVF failure rate. However, insufficient vessel management prior to the AVF procedure, as well as its use for intravenous therapy or blood tests, can result in vein damage and stenosis that may be difficult to detect during DUS mapping [17]. Patients with underlying conditions at risk for pulmonary hypertension require a distal AVF to prevent acute pulmonary edema. Secondary pulmonary hypertension following AVF creation did not increase the risk of AVF failure in maintenance hemodialysis patients, but significantly raised the mortality risk for this subset of patients [18]. Using small vessels for distal anastomosis in these cases can reduce the risk of congestion [19]. The relationship between arterial diameter and AVF outcome was studied on radio-cephalic fistulas. Immediate failure, on the same day of procedure, and early failure, during the first 8–12 weeks after surgery, are very frequent when small caliber arteries are chosen for afferent vessel (<1.5–1.6 mm). Malovrh reports an immediate failure of 55% and an early failure of 64% for arteries with diameter ≤ 1.5 mm compared to much lower rates, 8% and 17%, respectively, for arteries with diameter >1.5 mm. Parmar et al. report a 46% early failure rate for arteries with diameters < 1.5 mm compared to no failure for diameters > 1.5 mm [20,21]. Calcifications are identified ultrasonographically by the presence of hyperechogenicity in the arterial wall and irregularities in the intimal layer. While these alterations are easily recognizable, quantifying them proves challenging. Despite not being a contraindication to fistula surgery, they can still impact the outcome and may pose surgical challenges [22]. The pre-operative venous DUS involves the evaluation of the superficial and deep venous system of the upper limb, from the wrist to the central veins; for the latter, in particular, the ultrasound examination allows an easy evaluation up to the most distal portion of the subclavian vein. When there is a change in blood flow in the deep vein, it is important to consider the possibility of deep vein thrombosis in the proximal portion of the subclavian or innominate vein, which may not always be easily visualized with Doppler ultrasound. In these instances, an angio CT scan or alternatively a venography can be beneficial for further evaluation [23,24]. The diameter of the vein should be assessed at multiple points along the upper limb, using both longitudinal and transverse measurements. It is important to avoid applying excessive pressure with the probe in order to ensure an accurate reading and prevent underestimation of this measurement. Based on their results, Silva et al. suggest, in the presence of a tourniquet, a minimum venous diameter ≥ 2.5 mm while, in the absence of a tourniquet, Mendes et al. suggest a minimum venous diameter > 2 mm [25,26]. Even for the veins of the arm there are no well-documented indications on the minimum venous diameter but a value of at least 3 mm is still recommended [20]. After execution of the AVF, the vein tends to dilate under the influence of the increased blood flow. This ability of the vein to dilate is called “venous distensibility” and can be assessed during the pre-operative mapping by measuring the venous diameter before positioning a tourniquet for at least 2 min and after and evaluating the percentage increase [20]. The impact of venous distensibility on fistula outcomes was examined in two studies. Malovrh’s research found that higher venous distensibility predicts immediate AVF failure, with successfully packed fistulas showing a 48% dilation rate compared to 11% in failed fistulas. Other authors have suggested a link between the presence and size of collateral vessels and AVF maturation failure. Wong et al. emphasized the negative effect of collateral vessels within 5 cm of the anastomosis site on fistula functionality, while Beathard et al. stressed the significance of larger collateral vessels leading to more frequent AVF maturation failures [27,28]. As previously mentioned, it is evident that planning the placement of a fistula should consider different anatomical and clinical factors. Bozzetto et al. have conducted a notable study using a computational model to forecast the amount of blood flow that can be attained post AVF maturation. These data could assist in the surgical planning of AVFs, limiting those with inadequate or excessive blood flow, and consequently enhancing AVF patency rates and the overall clinical outcome of renal replacement therapy [29].

The lack of maturation is a relevant drawback of native AVF with inadequate flow and diameter increase to support hemodialysis. Since the introduction of the “native AVF wherever possible concept”, the incidence of native AVF failure increase was possibly related to the use of inadequate conduits as small marginal veins. An AVF is deemed mature when there is a flow rate of 600 mL/min, a vein diameter of 6 mm, less than 6 mm deep, and can be cannulated for dialysis within 6 weeks [30,31]. The exact mechanism of maturation is unclear, while advancements in assisted maturation techniques and an understanding of the underlying physiology in AVF development will play a role in improved AVF maturation and survival [32,33]. Following the creation of an AVF, a notable number of patients have been reported to experience a lack of maturation. This complication is mainly attributed to technical defects at the AVF anastomosis site, with arterial or venous issues such as unrecognized small diameters or stenosis being the most common culprits. Fortunately, these challenges can generally be addressed through surgical or endovascular interventions [34]. The percutaneous transluminal angioplasty (PTA), defined as the dilation of immature AVFs using an inflatable balloon catheter, is the most employed tool to overcome vein stenotic defects with acceptable outcomes in the current literature. Chang et al. demonstrated that AVF stenoses that had been treated with PTA had higher smooth muscle proliferative indices than those from de novo stenotic lesions [35]. Moreover, a recent large multi-center randomized control trial demonstrated a significantly higher primary patency rate at 6 months in favor of the DEB use group (81.4%) when compared with the standard balloon group (59%) [36]. A restenosis is possible a long time after execution of AVF. These outcomes are partially or entirely attributable to neointimal hyperplasia (NIH), which represents the vessel’s healing response to the trauma caused by endovascular interventions. In a retrospective study, Swinnen et al. demonstrated positive results in the treatment of restenosis in arteriovenous fistulas with drug-eluting balloons, significantly reducing the need for re-interventions [37]. 

The procedure proposed in this study is a modified version of the BAM technique, incorporating DEB and primarily used in patients at risk for FAV non-maturation or in cases of unsatisfactory AVF appearance during the initial procedure. Aligned with the National Kidney Foundation Disease Outcomes Quality Initiative, the intraoperative DUS-guided BAM with DEB aims to enhance AVF maturation through targeted intervention [38,39].

BAM is generally reported as a tool to improve the AVF flow after a failed maturation. In the reported experience we have employed different intraoperative signs to identify AVF at risk of no maturation, including thrill absence, thrill reduction at 1 cm proximal to the AVF anastomosis, and a reduced DUS volume flow (<200 mL/min) [40].

To avoid multiple procedures and reduce AVF maturation time, the intraoperative DUS-guided BAM with DEB has been used. When used after AVF failure, this technique generally consists of efferent vein PTA through transarterial access under fluoroscopy guidance. In our experience, all procedures were performed during the AVF creation through a direct puncture of the efferent vein. Under DUS guidance, PTA was effectively controlled without the need for contrast agents or radiation. Follow-up assessments at the venipuncture site did not reveal any cases of stenosis. The use of BAM has also been documented in the literature for fistulas involving small vessels. Traditionally, veins smaller than 3 mm in diameter were not considered suitable for AVF creation due to low maturation rates. However, studies by Wang et al. and Voto et al. suggest that veins smaller than 2.5 mm can successfully support functional AVFs in over 75% of patients using BAM prior to anastomosis, negating the need for a sheath. This technique can preserve proximal veins for future use, which is particularly important for younger patients requiring multiple AVFs. BAM also appears to eliminate the preference for more proximal access in cases of small vessels. Although vein size remains a key factor in AVF success, BAM has shown promise as a complementary technique [41,42]. 

The BAM method continues to stir up controversy within the medical community as it is debated whether it truly improves and speeds up the development of AVF maturation. There are studies that support the efficacy of the technique, while others argue against its use. However, the majority of studies support the use of BAM, as detailed in Table 2. DerDerian et al. note that significant complications following BAM are not observed at a concerning frequency. Although complications may be more common in BAM procedures performed in the forearm with larger balloons, each complication occurs in less than 10% of the procedures [16].

The most recent literature showed how the BAM procedure can significantly increase the maturation rate and the secondary patency, mainly when PTA is performed with DEB [35,43,44,45]. Combining BAM with local therapeutic application of an anti-stenotic agent may lead to the creation of mature AVFs in patients with small vessels, potentially eliminating the need for multiple post-maturation angioplasties and creating a beneficial outcome for all involved [46]. 

A consistent contribution to the development of DUS PTA comes from the study of Wakabayashi et al., reporting 4869 cases of fistula stenosis treated with the use of DUS. The major lesions in outflow vein stenoses were located around the anastomosis in 1493 cases and in the intermediate region in 251 cases. Early success was obtained in 4731 of 4869 cases (97.2%). In stenoses cases, early success was achieved in 4288 of 4414 cases (97.1%). The primary patency rate was 94.3% at 1 month [47]. Other authors reported early success in 93% of 228 procedures in 140 patients [48]. Thus, more patients can be given the opportunity to undergo the creation of an AVF for long-term hemodialysis as confirmed by the experience of De Marco Garcia LP et al., where the use of intraoperative BAM allowed the creation of AVF with suboptimal veins that otherwise would not have been used with maturation rate >96% [49,50]. The main clinical benefits of BAM with primary balloon angioplasty, therefore, are that it could significantly reduce the duration and number of tunneled dialysis catheter; and allow for the development of a large diameter and more fibrous AVF (due to repeated episodes of injury followed by healing), which might be easier to cannulate and also be more resistant to cannulation injury, with a potential requirement for a lesser degree of cannulation skills [46,47,51]. BAM should be considered as a useful tool to improve the primary patency of a fistula and reduce the reoperation rate. Furthermore, to improve the long-term patency of an AVF is a must, since the changes in hemodynamics and vascular structures do not necessarily allow the creation of a new AVF [52,53].

**Table 2 biomedicines-12-01005-t002:** Experiences of BAM reported in the literature.

References	N Patients Treated	PTA with DEB	% of Technical Success
Li et al. [10]	148	No	78%
Lookstein et al. [35]	330	Yes (170)	82%
		No (160)	59%
Wang et al. [41]	254	No	100%
Voto et al. [42]	662	No	100%
Irani et al. [44]	119	Yes (59)	81%
		No (60)	61%
Huang et al. [48]	290	No	94%
De Marco Garcia et al. [50]	62	No	85%
Kao et al. [54]	38	No	95%
Kohiyama et al. [55]	205	No	100%
Kanchanasuttirak et al. [56]	1427	No	97%
Kim et al. [57]	92	No	90%
Elkassaby et al. [58]	153	No	78%
Rizvi et al. [59]	54	No	55%
Gallagher et al. [60]	45	No	99%

This study is subject to various limitations which included the need for further studies to ensure accurate comparisons and appropriate group selection. Additionally, the outcomes may have been influenced by the surgeons’ individual decisions regarding BAM despite the use of DUS criteria. Due to its small sample size, this study was constrained in its ability to draw definitive conclusions. Further research with larger population sizes and standardized protocols is necessary to validate these findings.

## 5. Conclusions

Our experience has shown that identifying early signs of AVF failure during surgery greatly influences the decision to perform additional procedures. Intraoperative BAM proved to be a safe and effective adjunctive procedure, with minimal complications. It also led to faster AVF maturation times compared to the 4-week average reported in the literature. Increasing the use of intraoperative BAM could potentially reduce the need for reinterventions in AVF patients, particularly in cases where small veins are involved. While short-term challenges may arise, the long-term benefits of autogenous access include a decreased risk of access-related infections and improved patency rates. 

## Figures and Tables

**Figure 1 biomedicines-12-01005-f001:**
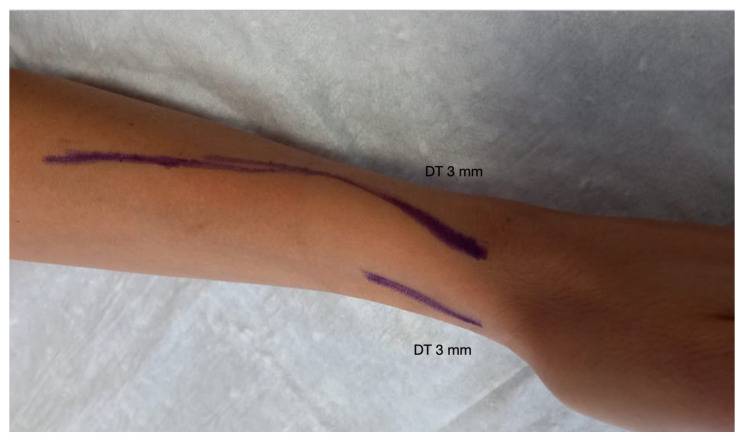
Pre-procedure mapping.

**Figure 2 biomedicines-12-01005-f002:**
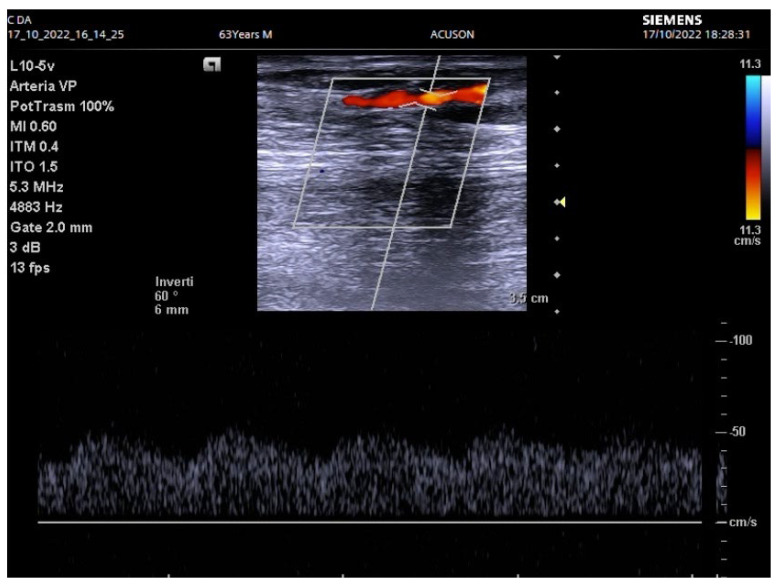
Low flow density in efferent vein with stenosis.

**Figure 3 biomedicines-12-01005-f003:**
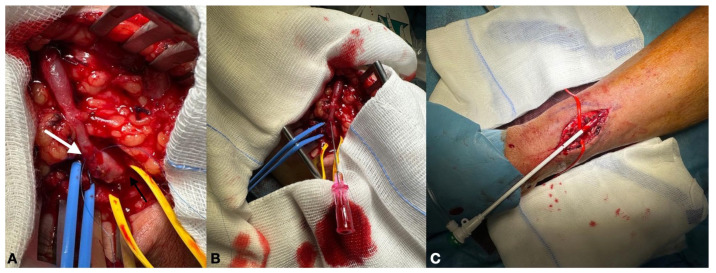
(**A**,**B**) Vascular access from the efferent vein (white arrow), 1 cm after the anastomosis (black arrow); (**C**) with 5F introducer radial sheath in a retrograde fashion.

**Figure 4 biomedicines-12-01005-f004:**
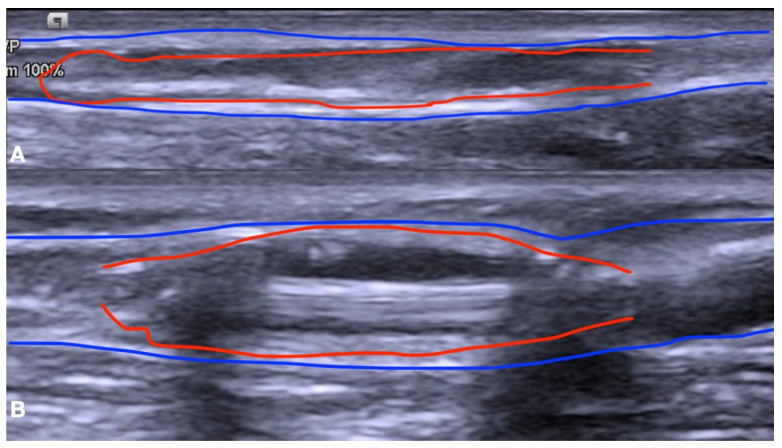
DEB positioned at the level of the lesion (**A**) and after achieving of nominal pressure (**B**). The blue lines indicate the outline of the vein while the red lines delineate the edges of the balloon.

**Figure 5 biomedicines-12-01005-f005:**
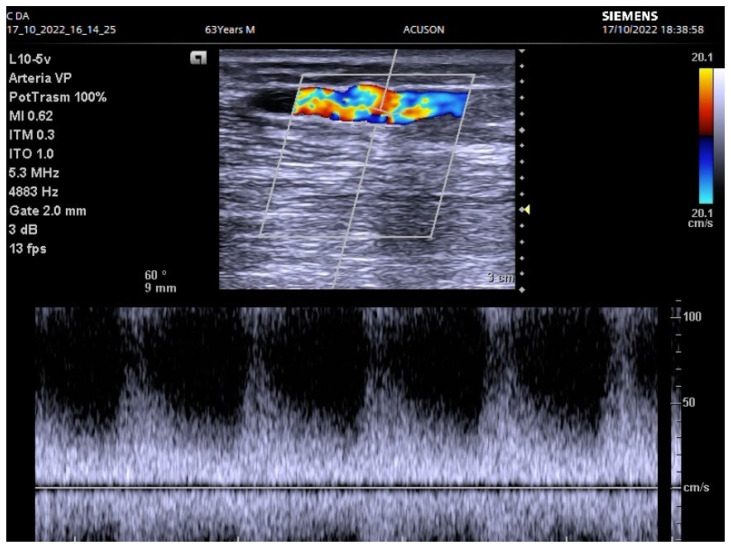
Efferent vein flow after PTA.

**Table 1 biomedicines-12-01005-t001:** Non-anatomic patient variables.

		N %
Diabetes	None	5 (16.1)
	Not requiring insulin	9 (29)
	Controlled by insulin	15 (48.4)
	Type 1 or uncontrolled	2 (6.5)
Tobacco Use	None (>10 years ago)	4 (12.9)
	Quit 1–10 years ago	11 (35.5)
	Current within last year, <1 package per day	9 (29)
	Current within last year, >1 package per day	7 (22.6)
Hypertension	None	0
	Controlled with 1 drug	8 (25.8)
	Controlled with 2 drugs	21 (67.7)
	Requiring > 2 drugs or uncontrolled	2 (6.5)
Cardiac status	Asymptomatic, with normal electrocardiogram	16 (51.6)
	Asymptomatic, but with remote myocardial infarction by history (6 months) or occult myocardial infarction	13 (41.9)
	Any one of the following: stable angina, no angina but significant reversible perfusion defect on dipyridamole thallium scan, significant silent ischemia (1% of time) on Holter monitoring, ejection fraction 25% to 45%, controlled ectopy or asymptomatic arrhythmia, or history of congestive heart failure that is now well compensated	2 (6.5)
	Any one of the following: unstable angina, symptomatic or poorly compensated or recurrent congestive heart failure, ejection fraction < 25%, myocardial infarction ≤ 6 months	0
Pulmonary status	Normal	15 (48.4)
	Asymptomatic or mild dyspnea	10 (32.2)
	Between normal and asymptomatic or mild dyspnea	6 (19.4)
	Vital capacity less than 1.85 L, FEV1 < 1.2 L or <35% of predicted, maximal voluntary ventilation < 50% of predicted, PCO2 > 45 mm Hg, supplemental oxygen use medically necessary, or pulmonary hypertension	0
Functional status	No impairment	23 (74.2)
	Impaired, but able to carry out ADL without assistance	7 (22.6)
	Needs some assistance to carry out ADL or ambulatory assistance	1 (3.2)
	Requiring total assistance for ADL or non-ambulatory	0

ADL, Activities of daily living; FEV1, forced expiratory volume in 1 s; PCO2, partial pressure of carbon dioxide.

## Data Availability

In appropriate cases, it can be requested from the corresponding author.

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
