# Peer review of "Improved Ultrasound-Guided Balloon-Assisted Maturation Angioplasty Using Drug-Eluting Balloons in the First Autogenous Arteriovenous Fistula Procedure: Early Experience"

_biomedicines, 2024, doi:10.3390/biomedicines12051005_

Round 1
Reviewer 1 Report
Comments and Suggestions for Authors
1. Methods: How was the 5fr access site managed after angioplasty? Was this closed with direct manual pressure or suture repair?
2. Results: What balloon lengths and diameters were used? Was the entire vein treated or just specific areas of stenosis?
3. Results: Do the authors have a control group of patients in whom they did not perform the BAM procedure? What was their maturation rate?
4. Discussion: For patients with small veins, why not do the angioplasty prior to performing the anastomosis through the open vein end? This would avoid having to place the 5fr sheath.
Comments on the Quality of English Languagenumerous grammatical and spelling mistakes present
Reviewer 2 Report
Comments and Suggestions for Authors
Dear authors I want to congrats for your study. I want a table in the discussion section with main articles with a significant number of cases regarding the case presented. In the same section I would like you to compare your study with others in the literature. Please modify the conclusions to be more clear and concise. There is nothing to look for bibliographic references to conclusions.
Round 2
Reviewer 2 Report
Comments and Suggestions for Authors
I agree your modification of the article.